# Systematic Identification of the RNA-Binding Protein STAU2 as a Key Regulator of Pancreatic Adenocarcinoma

**DOI:** 10.3390/cancers14153629

**Published:** 2022-07-26

**Authors:** Xiao Wang, Wenbin Kuang, Jiayu Ding, Jiaxing Li, Minghui Ji, Weijiao Chen, Hao Shen, Zhongrui Shi, Dawei Wang, Liping Wang, Peng Yang

**Affiliations:** 1State Key Laboratory of Natural Medicines of China Pharmaceutical University, Jiangsu Key Laboratory of Drug Design and Optimization of China Pharmaceutical University, Nanjing 210009, China; 1520220107@cpu.edu.cn (W.K.); 3320010996@stu.cpu.edu.cn (J.D.); 3219010060@stu.cpu.edu.cn (J.L.); 3220010055@stu.cpu.edu.cn (M.J.); 3220010033@stu.cpu.edu.cn (W.C.); 3221010133@stu.cpu.edu.cn (H.S.); 3221010085@stu.cpu.edu.cn (Z.S.); 3321010337@stu.cpu.edu.cn (D.W.); 1720180104@stu.cpu.edu.cn (L.W.); 2Department of Medicinal Chemistry, School of Pharmacy, China Pharmaceutical University, Nanjing 211198, China

**Keywords:** pancreatic adenocarcinoma, RNA-binding protein, STAU2, prognosis

## Abstract

**Simple Summary:**

Pancreatic adenocarcinoma (PAAD) is one of the most common tumors of the gastrointestinal tract and is difficult to diagnose and treat due to tumor heterogeneity and the immunosuppressive tumor microenvironment. RNA-binding proteins have been studied and their dysregulation has been found to play a key role in altering RNA metabolism in various malignancies. STAU2 is one of them. To investigate the role of STAU2 in PAAD, we monitored the signaling pathway by regulating substrate mRNA and experimentally confirmed that STAU2 is the most potential biomarker for the occurrence and development of PAAD. Furthermore, we found that high expression of STAU2 not only contributes to immune evasion but also correlates with sensitivity to chemotherapeutic agents, suggesting that STAU2 may be a potential target for combined natural therapy. These results demonstrate that STAU2 is a novel prognostic and diagnostic biomarker for PAAD, revealing STAU2′s utility in cancer therapy and drug development.

**Abstract:**

Pancreatic adenocarcinoma (PAAD) is a highly aggressive cancer. RNA-binding proteins (RBPs) regulate highly dynamic post-transcriptional processes and perform very important biological functions. Although over 1900 RBPs have been identified, most are considered markers of tumor progression, and further information on their general role in PAAD is not known. Here, we report a bioinformatics analysis that identified five hub RBPs and produced a high-value prognostic model based on The Cancer Genome Atlas (TCGA) and Genotype-Tissue Expression (GTEx) datasets. Among these, the prognostic signature of the double-stranded RNA binding protein Staufen double-stranded RNA (*STAU2*) was identified. Firstly, we found that it is a highly expressed critical regulator of PAAD associated with poor clinical outcomes. Accordingly, the knockdown of *STAU2* led to a profound decrease in PAAD cell growth, migration, and invasion and induced apoptosis of PAAD cells. Furthermore, through multiple omics analyses, we identified the key target genes of *STAU2*: Palladin cytoskeletal associated protein (*PALLD*), Heterogeneous nuclear ribonucleoprotein U (*HNRNPU*), SERPINE1 mRNA Binding Protein 1 (*SERBP1*), and DEAD-box polypeptide 3, X-Linked (*DDX3X*). Finally, we found that a high expression level of *STAU2* not only helps PAAD evade the immune response but is also related to chemotherapy drug sensitivity, which implies that *STAU2* could serve as a potential target for combinatorial therapy. These findings uncovered a novel role for *STAU2* in PAAD aggression and resistance, suggesting that it probably represents a novel therapeutic and drug development target.

## 1. Introduction

Pancreatic adenocarcinoma (PAAD) is the most fatal of any common solid-malignancy cancer [1,2]. In 2020, the survival rate was poor, and the 5-year rate is 9.2%. PAAD is considered to be the third primary cause of cancer-related mortality in the Western world [3,4]. In the United States, an estimated 47,050 deaths are expected. By 2030, it will be the second leading cause, particularly given persistently rising incidence and a minimal change in mortality rates [5,6]. PAAD develops from the following three best-characterized precursor lesions: pancreatic intraepithelial neoplasia (PanIN), intraductal papillary mucinous neoplasms (IPMN), and mucinous cystic neoplasms (MCN) [7,8]. Despite immense gains in the molecular understanding of PAAD, early diagnosis and prognosis remain very poor. At present, there is no targeted therapy that works better than chemotherapy, and the only available treatment is surgery, which is only available to a small number of patients with resectable tumors [9]. Therefore, a global understanding of the intricate pathogenesis of PAAD and the development of new prognostic biomarkers and drug targets are of critical importance for improving therapeutic strategies and survival rates.

High-throughput analyses of whole tumor cell transcripts suggest that RNA-based mutations play critical roles in cancer pathogenesis and the development of tumor heterogeneity [10,11]. Tumors are invariably associated with faulty RNA metabolism that disrupts the homeostasis of protein isoforms via oncogenic or tumor-suppressor signaling pathways. Besides somatic mutations in the target genes, dysregulation of RNA-binding proteins (RBPs) alters RNA metabolism in a variety of malignancies [12,13]. RBPs directly bind to numerous classes of RNAs to form dynamic ribonucleoprotein (RNP) complexes, which modulate all biochemical aspects of RNA life, including maturation, modification, splicing, transport, localization, decay, and translation [14,15]. Thus, dysregulation of RBPs results in transcriptomic and proteomic changes in tumor cells, which in turn affect cell growth, proliferation, invasion, and apoptosis [16,17]. Due to advances in screening techniques, more than 1900 human RBPs have been identified to date [18,19]. However, the role of RBPs in tumors is not fully understood.

Increasing pieces of evidence have proven that dysregulation of RNA metabolism by altering RBP expression is associated with PAAD onset and aggressiveness. For example, a member of the PCBP family protein, PCBP3, which increases the survival of PAAD cells, was regarded as a prognostic marker for PAAD [20]. SRSF1 and PTBP1 facilitate pancreatic cancer initiation and progression through alternative splicing regulation [21,22]. HuR, a member of the ELAV RBP family, not only affects mRNAs containing AREs in its 3′-UTRs, but also directly binds to miRNAs, thereby enhancing PAAD cell survival [16,23,24,25]. The RBP ZEB1 regulates epithelial-to-mesenchymal transition in PAAD cells by changing EMT-associated transcript expression [26]. ADAR1 regulates c-Myc stability through AKT signaling, thereby promoting PAAD growth [27]. Of particular interest are recent studies that have identified a role for RBPs in the immunotherapy of various types of cancer [28]. For example, eIF4E RBPs promote PD-L1 translation in mouse tumors, while eIF4E phosphorylation inhibitors disable PD-L1 translation [29]. MEX3B has been shown to affect immune resistance by disrupting HLA-A mRNA in cancer cells [30]. Therefore, in-depth investigation of the regulatory mechanisms of RBPs may be promising for developing innovative immunotherapy targets for pancreatic adenocarcinoma. Generally, these studies highlight the correlation of RBP dysregulation for PAAD tumorigenesis and progression. Furthermore, there is still an urgent need to systematically elucidate the overall functions of RBPs in PAAD prognosis and the immune microenvironment.

Herein, we used bioinformatic techniques in PAAD–TCGA and GTEx datasets to carry out a risk model based on five prognostic hub RBPs. Subsequently, based on hazard ratios with prognostic values and survival tests of these five hub genes, we focused on *STAU2* to evaluate the reliability of the RBP’s related signature. *STAU2* is a paralog of *STAU1*, which mediates a translation-dependent mRNA decay pathway (SMD) that is involved in multiple cellular processes [31,32,33]. Furthermore, several recent articles have reported that *STAU2* emerged as a critical mediator in tumor progression [34,35]. Nevertheless, the influence of *STAU2* on PAAD has not been investigated. Accordingly, our findings demonstrated that *STAU2* is abundantly expressed in PAAD, and downregulation of *STAU2* can significantly reduce the growth, invasion, and migration abilities of PAAD cells and induce apoptosis. Moreover, high *STAU2* expression was correlated positively with PAAD cell infiltration and immune checkpoint expression. On the flip side, high *STAU2* patients were more sensitive to the chemotherapy drugs (5-Fluorouracil and Gemcitabine) but more resistant to Erlotinib, an EGFR inhibitor. Collectively, based on the data from this study, we constructed and verified an RBP-based prognostic risk model that showed potential clinical application. Moreover, we demonstrated that *STAU2* is a novel regulator of PAAD initiation and progression, which could serve as a potential diagnostic and prognostic biomarker for combinatorial therapy to improve PAAD survival.

## 2. Materials and Method

### 2.1. Data Collection and Processing

For 170 TCGA–PAAD patients, gene expression files containing the number of untreated genes and transcript fragments in kilobases (FPKM) were obtained from Genomic Data Commons (GDC) using “TCGAbiolinks R”. The gene expression matrix of normal pancreas across 360 healthy donors was obtained from Genotype-Tissue Expression (GTEx). The gene list containing 1500 RNA binding proteins (RBPs) was obtained from the previous report. Using DESeq2 to analyze differentially expressed genes (DEGs) between TCGA–PAAD patients and GTEx-normal pancreatic tissue, adjusted *p*-values < 0.05 and 1.5-fold changes were considered statistically significant differences.

### 2.2. Identification of RBP Signature with Prognostic Significance

Using RBP genes that were significantly differentially expressed in pancreatic tumors and normal tissues, we performed univariate Cox regression analysis to isolate prognostic RBP genes. In detail, log-rank *p*-values were calculated to estimate the significance, and the survival plot was obtained through the “survminer” R package. These prognostically relevant RBPs were ranked based on the smallest absolute shrinkage and selection operator (LASSO). Further multiple stepwise Cox regression was performed to identify the hub RBPs with prognostic significance. Risk scores for each patient were calculated as follows:$$Risk\;score=\overset{n}{\underset{k=1}{\sum }}exp^{k}coef^{k}$$Here, “*n*” is the number of prognostic RBP genes (*n* = 6), “*exp^k^*” is the expression levels of the gene *k*, and “*coef^k^*” is the estimated coefficient value of gene k in multivariate Cox regression analysis. After normalizing the gene expression levels (FPKM) using the estimated regression coefficient, we calculate the weighted risk scores of selected hub RBPs. Ultimately, we assigned 170 PAAD patients to the low-risk group (*n* = 85) and high-risk group (*n* = 85), regarding the median risk scores as the threshold point.

### 2.3. Identification of Differentially Expressed Genes between Risk Subgroups

A total of 170 patients with PAAD were divided into high and low risk subgroups according to risk scores, and differential gene expression analysis (DGE) was performed on the two subgroups using DESeq2. Adjusted FDR < 0.05 and |log2 (fold change) | > 1 were used as the threshold of significance.

### 2.4. Gene Set Enrichment Analysis

DEGs were tested for functional enrichment using the “clusterProfiler” R package based on the GO (Gene Ontology) and KEGG Pathway databases, respectively. Gene set enrichment analysis (GSEA) was used to detect abnormal signaling pathways assigned to hallmark gene sets in high-risk subgroups.

### 2.5. Combination Analysis of Gene Expression, DNA Methylation, and Genetic Alternation in PAAD Patients

DNA methylation array (Illumina Human Methylation 450, San Diego, CA, USA) and copy number alternation data were downloaded from the GDC data portal using the TCGAbiolinks R package. To explore the upstream regulation of prognostic RBPs, we calculated the correlation coefficients of RBP genes’ FPKM with DNA methylation levels in the promoter region of RBPs and log2-transformed copy numbers in RBPs, respectively.

### 2.6. Validation of Prognostic RBP Signature

Microarray data, somatic mutation, and genetic alternation information across 461 PAAD patients were obtained from the international cancer genome consortium (ICGC) data port (https://dcc.icgc.org/, accessed on 3 February 2021). Cox one-way regression and Covar multiple regression analyses were used to validate the predicted value of the RBP attribute of the *STAU2* node.

### 2.7. STAU2 Genetic Alternation Analysis

The genetic variation signature of the *STAU2* gene was obtained from the cBioPortal website (https://www.cbioportal.org/, accessed on 3 February 2021), which summarizes its variation frequency, mutation type and copy number (CNA) changes. The comparison of genetic alteration characteristics of *STAU2* across multiple TCGA tumors was performed using the “Cancer Type Summary” module. The “Mutation” module in the cBioPortal website was used to construct the schematic diagram of the structure of mutated *STAU2*. Kaplan–Meier plots with log-rank *p*-values were plotted to show the relationship between *STAU2* genetic alteration and the overall survival of TCGA tumors.

### 2.8. Prediction of Diagnostic Effect of ROC Curve on STAU2

The calculated ROC curves and AUC values were analyzed with the R pROC package, as shown in ggplot2. AUC values between 0.5 and 0.7 indicate model success.

### 2.9. Analysis of Differentially Expressed Genes with STAU2

Genes associated with *STAU2* expression in PAAD were detected using Linkedomics (http://www.linkedomics.org/login.php, accessed on 3 February 2021). Volcano plots were used to visualize, and they filtered regulated and decreased genes separately. The significant correlation between genes was evaluated by Pearson’s test.

### 2.10. Protein–Protein Interaction (PPI) Network Construction

We used the STRING database to predict protein–protein interactions (PPIs) and create PPI networks. The STAU2 protein–protein interaction network was then proposed, and two major closely interacting proteomes were observed. It was found that 50 proteins were closely related to STAU2. Then, KEGG enrichment analysis was performed on them, and 20 of them were used to construct the network graph.

### 2.11. Identification of STAU2-Target Genes

STAU2-CLIP data were downloaded from GSE134971. Narrow peaks of two STAU2-CLIP samples were merged using IDR software. The RNA sequence with STAU2-CLIP peaks was summited to MEME to identify putative STAU2-binding motifs. Then, STAU2-CLIP peaks were annotated to the nearest gene using the “annotatePeak” function of the “ChIPseeker” R package. The host genes of STAU2-CLIP peaks were regarded as STAU2-binding genes.

The correlation between *STAU2* and other protein-coding genes was analyzed across 170 TCGA–PAAD patients using the FPKM matrix. The *STAU2*-associated gene was defined as a gene with |R coefficient| > 0.3.

### 2.12. Analysis of Immune-Related Information of STAU2

The correlation between immune cell infiltration and *STAU2* expression was obtained using the TIMER online website, and Pearson’s correlation analysis was used to obtain the correlation. Correlation of *STAU2* with immune cell gene markers in pancreatic adenocarcinoma was determined using timer analysis.

To reliably assess immune scores, we used immunodeconv, an R package that uses the built-in xCell algorithm. The generated heatmaps are implemented by the R (v4.0.3) ggplot2 and pheatmap packages.

The R software package “circle” was used to plot Chord diagram. The correlation between the expression of *STAU2* gene and eight immune checkpoints was analyzed by Spearman analysis, and the correlation coefficient and significant correlation were gained. Utilizing the TIMER2 web server (http://timer.cistrome.org/, accessed on 14 October 2021), we obtained the immune infiltration scores of cancer-related fibroblasts, mast cells, endothelial cells, cancer-associated fibroblast, CD8+ T-cells, CD8+ central memory T-cells, CD4+ memory T-cells, memory CD4+ central memory T-cells, Th1 CD4+ T-cells, NK T-cell, plasma B-cells, the common lymphoid progenitor, and the granulocyte−monocyte progenitor. Pearson correlation analysis was used to show the association between *STAU2* expression levels and improved immune system infiltration in PAAD patients.

### 2.13. Kaplan–Meier Plotter Database Analysis

To analyze the prognostic value of *STAU2* in PAAD, we used a web-based gene expression database and survival information from the KM Plotter. Patient samples were divided into two groups based on median expression (high and low expression) and hazard ratios (HR) with 95% confidence intervals (95% CIs), and logrank *p*-values were used to analyze the hazard ratios (HR) of immune cell subsets.

### 2.14. Drug Sensitivity Analysis of STAU2 in Pancreatic Adenocarcinoma

Using the Gene Set Cancer Analysis database (GSCA) online website and according to the data from Genomics of Drug Sensitivity in Cancer (GDSC, https://www.cancerrxgene.org/, accessed on 14 October 2021) and Cancer Therapeutics Response Portal (CTRP, https://portals.broadinstitute.org/ctrp/, accessed on 14 October 2021), the 30 drugs with the most sensitivity and predictive accuracy against *STAU2* were predicted. For IC_50_ and gene expression correlation analysis of first-line drugs for pancreatic adenocarcinoma, tumor RNA-seq (FPKM) data downloaded from GDC were used, converted to TPM format, and data were normalized to log2 (TPM + 1). Half-maximal inhibitory concentration (IC_50_) data for predicted samples were obtained from GDSC and CTRP, predictions were performed using a “pRRophetic” R package, and IC_50_ were estimated by comb regression.

### 2.15. Cell Culture

The PANC-1 cell line, BXPC-3 cell line (pancreatic cancer), and HPDE6-C7 cell line (normal pancreas cell) were obtained from the American Type Culture Collection Center (ATCC) and supplemented with 10% (FBS) in endotoxin-free in DMEM cultivated; BXPC-3 pancreatic adenocarcinoma cells were maintained in RPMI-1640 in 10% FBS. Penicillin streptomycin and plasma cytokine prophylactic were added to the medium.

All human cell lines have been authenticated using STR (or SNP) profiling within the last three years. All experiments were performed with mycoplasma-free cells.

### 2.16. Cell Proliferation Assays

Cell proliferations were evaluated using the Cell Counting Kit-8 (Share-bio, SB-CCK8, Shanghai, China). First, 5000 shNC-PANC-1 cells and sh*STAU2*-PANC-1 cells were seeded in each well of a 96-well plate for 5 days. CCK8 reagent was added to the wells of each dish daily. Plates of treated cells were incubated in an incubator for 4 h. The enzyme plate analyzer was used to detect the absorbance at 450 nm using a microplate reader (Bio-Tek SynergyH1, Vineland, NJ, USA). For cell growth assays, cell viabilities were determined at 0, 24, 48, 72, and 96 h. Three independent trials were performed. For antiproliferative activity of the compound, data were fitted in nonlinear regression, and IC_50_ values were calculated by GraphPad Prism 8.0.

### 2.17. Lentivirus Production and Infection

Lentivirus particles of short-hairpin RNA against *STAU2* (pGV112-sh*STAU2*) and its scrambled control (PGV112-shNC) were constructed and purchased from Genechem Co. Ltd. (Shanghai, China). Lentivirus-induced *STAU2* is transformed into PANC-1 cells to generate PANC-1 knockdown cells for *STAU2*. In general, the lentiviral particles were collected and transferred directly to PANC-1 cells after 72 h of transfection. Lentivirus-containing PANC-1 cell lines were inoculated at 32 °C, 1200 rpm for 90 min. After the rotation inoculation, puromycin was added to the cultured PANC-1 cells to select positively infected cells. The sh*STAU2* sequence was 5′-CCGGGCCAGGGAACTCCTTATGAATCTCGAGATTCATAAGGAGTTCCCTGGCTTTTTG-3′, and the non-targeting shRNA sequence was 5′-CCGGTTCTCCGAACGTGTCACGTTTCAAGAGAACGTGACACGTTCGGAGAATTTTTG-3′.

### 2.18. Real-Time Quantitative PCR (RT-qPCR)

The total RNA isolated from cells with RNA-easy reagent was reverse transcribed using Hiscript III 1st Strand cDNA Synthesis (Vazyme, R323-01, Nanjing, China). The RT-qPCR reaction was carried out using the ChamQ SYBR qPCR Master mix (Vazyme, Q331-02, Nanjing, China). See Appendix A for primers.

### 2.19. Western Blot Analysis

Cells were grown at 5 × 10^6^/mL in T-75 flasks. The treated cells were lysed with RIPA (Thermo Fisher Scientific, 89901, Waltham, MA, USA), a phosphatase inhibitor (Roche, 04906845001, Basel, Switzerland), and a protease inhibitor (Roche, 04693132001). Lysates were quantified and boiled in SDS sample buffer, then fractionated with SDS-Page and transferred to PVDF membranes. Blocking, antibody incubation, and washing were performed in buffer containing 0.05% (*v*/*v*) Tween-20 and 5% (*w*/*v*) non-fat dry milk. The primary antibody was diluted against the target protein in blocking solution. Membranes were incubated overnight with the primary antibodies listed in Appendix A. After four washes in blocking solution, spots were incubated with horseradish peroxidase-conjugated secondary antibody. Protein was detected by electrochemiluminescence. Finally, the Image J V1.8.0 software was used to perform protein quantification.

### 2.20. Colony Formation Assays

One thousand cells were plated in 24-well plates to analyze colony formation in dishes. Cells were fixed 2 weeks after pre-plating, stained with 0.1% crystal violet, and counted.

### 2.21. Migration Assays

In vitro cell migration assay of *STAU2* knockdown transfected PANC-1 cells or PANC-1 cells was performed using transwell plates (Falcon, 353097, New York, NY, USA). To begin, 200 μL DMEM without fetal bovine serum was inoculated on the upper surface at room temperature, and 500 μL DMEM medium containing 10% FBS was injected into the wells at a ratio of 2 × 10^4^ cells at room temperature. Cells were cultured for 24 h in transwell chambers, and cells that did not migrate to the upper surface were removed. Migrated cells were fixed on the underside with 4% paraformaldehyde and stained with 0.1% crystal violet for 5 min. The number of cells on the lower surface was counted under a microscope to determine the number of migration.

### 2.22. Invasion Assays

In vitro cell invasion assay of *STAU2* knockdown transfected PANC-1 cells or PANC-1 cells was performed using transwell plates. First, 200 μL DMEM without fetal bovine serum was inoculated on the upper surface of 20 μL Matrigel (Corning, 356234, New York, NY, USA) at room temperature, and 500 μL DMEM medium containing 10% FBS was injected into the wells at a ratio of 2 × 10^4^ at room temperature. Cells were cultured for 24 h in transwell chambers, and cells that did not migrate to the upper surface were removed. Infiltrated cells were fixed on the underside with 4% paraformaldehyde and stained with 0.1% crystal violet for 5 min. Cell penetration into the lower surface of Matrigel was counted, and the number of cells under the microscope was also counted.

### 2.23. Apoptosis Assays

Apoptosis was detected using YF^®^488-Annex V and PI Apoptosis Kit (US Everbright, Y6002, Suzhou, China). Cells were seeded at 1 × 10^6^/well, then washed twice with cold PBS and resuspended in 1× binding buffer at a concentration of 1 × 10^6^ cells/mL. Cell apoptosis under 488 nm excitation was detected by flow cytometry, and the results were analyzed using FlowJo V10 software.

### 2.24. RNA-Immunoprecipitation RT-qPCR (RIP RT-qPCR)

Cells were lysed in lysis buffer (100 mM KCl, 5 mM MgCl2, 10 mM Hepes pH 7.0, 1 mM DTT, 50 units/mL RNase out, 1× protease inhibitor cocktail, 1× PBS) at 4 °C for 2 h. A volume of 10% lysate was subjected to RNA isolation as input. Then, 10 mg Protein A-Agrose beads (Sigma-Aldrich, P1406-250MG, Darmstadt, Germany) were pre-treated with PBS three times and 2% BSA for 30 min and then incubated with 10 μL *STAU2* or IgG (as control) antibody at 4 °C for 2 h. Subsequently, the beads–antibody complex was incubated with cell lysate on rotation at 4 °C overnight. Beads were washed with PBS and then subjected to total RNA isolation. Purified RNA was reversely transcribed followed by RT-qPCR.

### 2.25. RNA-Seq and Data Analysis

Cells were harvested and total RNA was extracted using Beyozol Total RNA Extraction Reagent (Beyotime, R0011, Shanghai, China), following the instructions of the manufacturer. Total mRNA was enriched by Obligo(dT) beads then fragmented into short fragments and reversely transcribed into cDNA with random primers. After the second-strand cDNA was synthesized, cDNA fragments were purified, end repaired, poly(A) was added, and they were ligated to Illumina sequencing adapters. The ligation products were size selected by agarose gel electrophoresis, amplified by PCR, and sequenced using Illumina HiSeq2500 by Genedenovo Biotechnology Co., Ltd. (Guangzhou, China). Raw reads were cleaned to remove adapters or low-quality reads and rRNA mapped reads. Clean reads were mapped to the Homo sapiens’ genome. Gene expressions were quantified by FPKM (fragment per kilobase of transcript per million mapped reads) value and differential expression analysis was performed by R (v4.0.3) DESeq2 between two different groups, with false discovery rate (FDR) below 0.05 and absolute fold change ≥2 considered differentially expressed genes. Bioinformatic analysis was performed using the Omicsmart platform (https://www.omicsmart.com, accessed on 26 October 2021).

### 2.26. Statistical Analysis

Continuous variables between two groups were compared using Student’s *t*-test and Mann–Whitney Wilcoxon test. Kruskal Wallis One-Way ANOVA was used to examine differences in more than two groups. Analysis of variance (ANOVA) was used to test for variance between groups. The Benjamin–Hochberg method was used to adjust the false detection rate (FDR) of *p*-values for many comparisons. All statistical analyses were performed using R software (v 4.0.3). *p*-values < 0.05 were considered statistically significant.

## 3. Results

### 3.1. Identification of Prognostic RBP Signature

RNA-seq data from 170 PAAD specimens and 360 normal pancreas tissue samples were downloaded from the TCGA and GTEx cohorts. We preprocessed the data to identify differentially expressed RBPs using the limma software package. We screened out 477, downregulated 221, and upregulated 256 RBPs (Figure 1A). For a further selection of RBPs with the highest forecast value, using univariate Cox regression in a step-by-step analysis, we chose five hub prognostic RBPs and carried out a lasso regression to establish a risk model for PAAD patients (Figure 1B). Multiple stepwise Cox regression analyses through the TCGA cohort demonstrated that the mRNA levels of high-risk RBPs (*p* = 0.0312) and mutated KRAS genes (*p* = 0.01221) were independent prognostic factors (Appendix A). The expression profiles showed a marked overexpression of *STAU2*, *DDX60L*, *MRPS10*, *PARN*, and *TLR3* in PAAD samples compared to normal samples (Figure 1C). Moreover, the KM plot through the TCGA cohort demonstrated that PAAD patients with a higher risk score had poor overall survival, and high expression of *STAU2*, *DDX60L*, *MRPS10*, *PARN*, and *TLR3* was also associated with worse overall survival (Figure 1D, *p* < 0.01). Consistently, data from the international cancer genome consortium (ICGC) dataset showed similar results (Appendix A, *p* < 0.01), but no significant difference in *TLR3*. Collectively, our results showed that the five hub RBP-based risk model correlated closely with PAAD patients’ survival.

### 3.2. Transcriptomic Alternation in High-Risk PAAD Patients

The differentially expressed genes (DEGs) of different risk groups were tested in the TCGA–PAAD cohort to further analyze the underlying mechanism of the RBP genes: of 3918 DEGs, 3177 were downregulated and 741 upregulated (Figure 2A). Then, using GSEA and KEGG pathway analysis, Hallmark and KEGG functional enrichments were performed. They demonstrated that genes enriched in pancreatic secretion, E2F targets, G2M targets, MTORC1 signaling, and glycolysis signaling were significantly upregulated in the high-risk group, whereas in the low-risk group, genes enriched in MYOGENESIS, KARS signaling, and the pancreas’ beta-cell pathway were upregulated (Figure 2B,C). These results indicated that these RBPs might regulate PAAD occurrence and progression by affecting cellular proliferation and metabolic pathways.

### 3.3. Upstream Regulation of Alternated Hub RBPs in PAAD Patients

To explore the upstream regulation of these highly expressed RBPs, we examined the copy-number changes and methylation of five RBP genes based on the website server cBioPortal. The correlations were as follows: 0.42 between copy-number alterations with mRNA expression of *STAU2* (*p* < 0.001); −0.33 between DNA methylation with mRNA expression of *STAU2*, (*p* < 0.001) (Figure 3A); 0.093 between copy-number alterations with mRNA expression of *DDX60L* (*p* < 0.001); −0.44 between DNA methylation with mRNA expression of *DDX60L* (*p* < 0.001) (Figure 3B); 0.48 between copy-number alterations with mRNA expression of *MRPS10* (*p* < 0.001); −0.26 between DNA methylation with mRNA expression of *MRPS10* (*p* < 0.001) (Figure 3C); 0.032 between copy-number alterations with mRNA expression of *TLR3* (*p* < 0.001); −0.37 between DNA methylation with mRNA expression of *TLR3* (*p* < 0.001) (Figure 3D); 0.49 between copy-number alterations with mRNA expression of *PARN* (*p* < 0.001); and 0.3 between DNA methylation with mRNA expression of *PARN* (*p* < 0.001) (Figure 3E).

This result indicated that *STAU2* might be copying the number-regulating genes *MRPS10* and *PARN* and methylation-regulating genes *DDX60L* and *TLR3*.

### 3.4. High Expression of STAU2 Was Associated with a Negative Prognosis of PAAD

Among the five RBPs, *STAU2* had the highest hazard ratio, so we selected it for further analysis. We observed the status of *STAU2* gene modification based on the cBioPortal website sever. The frequency of the highest alteration in *STAU2* (>8%) was observed in patients with uterine Cowden syndrome (CS) tumors with “amplification” as the primary type. The amplification-type of *STAU2* was the main type of pancreatic cancer, and its mutation frequency is around 5% (Figure 4A). In addition, we investigated possible associations between *STAU2* mutations and PAAD clinical outcomes. The data in Figure 4B indicate that PAAD cases without genetically altered *STAU2* had a better overall survival prognosis (*p* = 0.0064) compared with cases with genetically altered *STAU2*, in which amplification is the most important part the alteration. These results are consistent with the poor prognosis of PAAD patients with *STAU2* amplification and high expression.

For patient specificity (geographic specificity) and clinical features specificity of STAU2, we respectively conducted expression and prognosis analysis in different conditions. For the geographical specificity of patients, we analyzed STAU2 expression in pancreatic cancer samples of different ethnicities (White, Asian, and Black) and found no significant differences. We also analyzed STAU2 expression in pancreatic cancer patients without and with radiation therapy and found that pancreatic cancer patients who received radiation therapy had significantly lower STAU2 expression than those who did not (Appendix A). A multivariate TCGA–Cox regression analysis showed that high levels of *STAU2* mRNA (*p* = 0.04562), age (*p* = 0.0458) and KRAS (*p* = 0.00519) were independent prognostic factors (Figure 4C). Analysis of a multivariate Cox regression through the ICGC dataset confirmed that a high *STAU2* mRNA level (*p* = 0.04703) is an independent prognostic factor (Appendix A). We also analyzed the influence of *STAU2* expression on the development of PAAD through the receiver operating curve (ROC), and the area-under-the-curve (AUC) value of *STAU2* was 0.96. These results suggested that *STAU2* expression might be associated with PAAD progression (Figure 4D).

We then analyzed *STAU2* mRNA levels in three pancreatic cell lines. Compared with the normal HPDE6-C7 cell line, PANC-1 and BXPC-3 cell lines showed a significant level of *STAU2* mRNA regulation (Figure 4E). Then, we analyzed the expression level of STAU2 protein in PAAD cell lines by Western Blot and found that the expression of STAU2 was higher in PANC-1 and BXPC-3 cells compared to HPDE6-C7 cells (Figure 4F).

### 3.5. Down-Regulation of STAU2 Significantly Reduces the Burden of PANC-1 Cell Line

We depleted *STAU2* in the PANC-1 cell line, which we selected for further experiments (Figure 5A). Invasion is an important part of migration, and cells with high migration ability generally have high an invasion ability. Invasion and migration ability are both markers for tumor malignancy. Deleting *STAU2* significantly inhibited cell cloning (Figure 5B), cell growth (Figure 5C), migration, and PANC-1 cell line invasion (Figure 5D). In addition, inhibition of *STAU2* in PANC-1 cells induced apoptosis (Figure 5E). WB results showed upregulation of E-cadherin, whereas N-cadherin, BCL-2, and caspase 7 were downregulated (Figure 5F). In summary, reducing the level of *STAU2* significantly reduced PAAD malignancy, suggesting that *STAU2* plays a key role in PAAD regulation and may serve as a potential target for novel anticancer drugs.

### 3.6. The Function Analysis of STAU2 and Its Related Genes

To better understand the mechanism behind *STAU2*, we divided pancreatic cancer samples collected by TCGA into high- and low-expressing *STAU2* groups to test the difference between them. Of the DEGs, 174 genes were downregulated and 104 upregulated (Figure 6A). Hallmark enrichment showed that upregulated genes were enriched in pancreatic beta cells, hedgehog signaling, UV-response, increased KRAS signaling, mitotic spindle, and the inflammatory response pathway. The downregulated genes were enriched in lower KRAS signaling, DNA repair, and oxidative phosphorylation (Figure 6B). According to the STRING interactive network, 50 proteins that can bind to *STAU2* were characterized (Figure 6C), and their 50 genes were enriched via the KEGG pathway on the online platform Omicsmart. The main pathways are RNA transport, spinocerebellar ataxia, mRNA surveillance, and ubiquitin-mediated proteolysis (Figure 6D). Then, we enriched the KEGG pathways with *STAU2*-related genes, which were then explored by the FPKM matrix for TCGA–PAAD patients (Appendix A). The main pathways are salmonella infection, ubiquitin-mediated proteolysis and endocytosis, spliceosome, and mRNA surveillance (Figure 6E). The dot plots show the hallmark pathway enrichment results of *STAU2*-related genes, which were found to be associated with mitotic spindle, protein secretion, MYC targets V1, UV-response, and TGF-β signaling (Figure 6F). These results indicated that *STAU2* and its related genes might regulate cell proliferation and the metabolism pathways of PAAD cells.

### 3.7. Identification of STAU2 Target Genes

To determine the downstream STAU2 target genes, we analyzed the genomic distribution of STAU2-CLIP peaks (GSE134971) [34] A strong enrichment of STAU2 binding to the last exon was noted on or close to the 3′ untranslated region (UTR) of mRNA (Figure 7A). As shown in Figure 7B, high-risk PAAD patients had an overrepresentation of the STAU2 binding motif. The logo representation plots show the putative STAU2 binding motifs overrepresented around *STAU2*-related genes (Figure 7C). TCGA patients were sorted according to *STAU2* expression from low to high. The heatmap shows the expression of *STAU2*-correlated genes (Appendix A). To investigate which gene expression could be regulated by *STAU2*, we performed a transcriptome-wide RNA-sequence analysis of STAU2-knockdown PANC-1 and control cells. We identified seven genes among the *STAU2*-correlated genes (TCGA–PAAD), STAU2-binding genes (GSE134971), and *STAU2*-correlated downregulated genes in sh*STAU2* samples (Appendix A and Figure 7D). Correlations between *STAU2* expression and *PALLD*, *HNRNPU*, *SERBP1*, *DDX3X*, *ALDH5A1*, *FAM8A1*, and *TBC1D5* expression levels were positive (Appendix A and Figure 7E). We then found that PAAD patients with high *PALLD*, *HNRNPU*, *SERBP1*, and *DDX3X* mRNA had a poor prognosis (Appendix A and Figure 7F). Moreover, using RT-qPCR, we confirmed the expression of these genes in shNC and sh*STAU2* PANC-1 cells, and the expressions of *PALLD*, *HNRNPU*, *SERBP1*, and *DDX3X* were markedly reduced (Figure 7G). We also confirmed that they were the direct editing targets of STAU2 (Appendix A and Figure 7H).

### 3.8. Correlation Analysis between STAU2 Expression and Immune Infiltration Cells

Immune cells play a critical role in the tumor microenvironment, and tumor infiltration is closely related to the onset, development, and metastasis of tumors. Herein, we analyzed the correlation between the expression of *STAU2* and six types of immune infiltration cells: B cells, CD4+ T cells, CD8+ T cells, neutrophils, macrophages, and dendritic cells, by the TIMER online website. The results showed that there was a positive correlation between the *STAU2* expression level and these infiltration cells, except for CD4+ T cells, for which there was no significant correlation (Figure 8A). To further investigate the effect of *STAU2* on different functional T cells, we analyzed the correlation between the expression of *STAU2* and tumor immune infiltration cells (TIICs) using the xCell algorithm in the PAAD dataset. The results showed that high *STAU2* expression correlates with a common lymphoid progenitor, granulocyte–monocyte progenitors, CD8+ T cell, CD8+ central memory T cell, and CD4+ memory T cell. Meanwhile, low *STAU2* expression was related to B plasma cells and CD4+ central memory and CD4+ Th1 T cells (Figure 8B and Appendix A). Moreover, we examined the relationship between the expression of *STAU2* and some immune checkpoint molecules: *SIGLEC15*, *TIGIT*, *PDCD1LG2*, *PDCD1*, *LAG3*, *HAVCR2*, *CTLA4*, and *CD274* (PD-L1). Interestingly, the expression of *STAU2* was significantly associated with them (Figure 8C). To further understand the crosstalk of *STAU2* in immune responses, we studied the correlation between *STAU2* expression in PAAD and various immunological characteristics. The genes were used to characterize immunosuppressive cells, including regulatory T cell (Treg), tumor-associating macrophage (TAM), and myeloid-derived suppressor cell (MDSC). Our finding also indicated that the expression of *STAU2* is positively related to these marker sets of immunosuppression cells through TIMER in the PAAD dataset (Figure 8D).

### 3.9. Predictive Analysis of STAU2 Expression in PAAD Using Immune System Cells

Since *STAU2* expression was significantly associated with immune system infiltration and poor prognosis, we further investigated whether it affects PAAD immune infiltration. The prognostic analysis was conducted based on expression in PAAD-related immune cell subsets. As shown in Figure 9A,B, high expression of *STAU2* and reduced infiltration of CD8+ T cells, eosinophils, macrophages, regulatory T cells, and type 2 T-helper cells were associated with a poor prognosis. However, under different levels of B cells, CD4+ memory cells, mesenchymal stem cells, and natural killer T cells, the expression of *STAU2* had no significant correlation with PAAD prognosis. These results suggested that the effect of *STAU2* may be partly due to immune infiltration.

### 3.10. Drug Sensitivity Analysis of the STAU2

We used the GDSC and CTRP databases to display the drug sensitivity of *STAU2* through bubble charts. Figure 10 shows that the increased expression of *STAU2* was associated with drug resistance. In the GDSC database, highly expressed *STAU2* might be sensitive to 17-AAG (Hps90 inhibitor) and RDEA119 and trametinib (MEK inhibitors). Conversely, highly expressed *STAU2* might be resistant to drugs such as BX-912 (PDK1 inhibitor), GSK1070916 (Aurora kinase inhibitor), and navitoclax (Bcl-2 inhibitor) (Figure 10A). In the CTRP database, high expression of *STAU2* might be resistant to BI-2536 (PLK inhibitor), GSK461364 (PLK inhibitor), and JQ-1 (BET inhibitor) (Figure 10B). Then, we explored the correlation between *STAU2* expression and the IC50 of the post-pancreatic surgery adjuvant chemotherapy drugs (5-Fluorouracil and Gemcitabine) and Erlotinib, used for targeted therapy for unresectable locally advanced pancreatic cancer. The result showed a negative correlation with the IC_50_ of 5-Fluorouracil and Gemcitabine, but a positive correlation with the IC_50_ of Erlotinib (Figure 10C). Then, we tested the IC_50_ values of the three drugs against shNC and sh*STAU2* in PANC-1 cells using cell counting kit 8 (CCK8). The result showed that 5-Fluorouracil and Gemcitabine increased the IC_50_ value of sh*STAU2* PANC-1 cells relative to shNC PANC-1 cells, whereas Erlotinib reduced the IC_50_ value (Figure 10D). These data suggest that patients with high levels of *STAU2* mRNA are more sensitive to 5-Fluorouracil and Gemcitabine but more resistant to Erlotinib, making these drugs potential targets for combination therapy.

## 4. Discussion

As the third-leading cause of cancer death in western countries [36], the diagnosis and treatment of PAAD patients is challenging due to tumor heterogeneity and the immunosuppressive tumor microenvironment [37]. Currently, surgery, chemotherapy, and radiotherapy are the main treatment strategies. However, side effects and drug resistance to chemotherapy and radiotherapy result in treatment failures [38]. Indeed, with a persistently rising morbidity and extremely poor prognosis, PAAD will become the second-leading cause of cancer death by 2030 [5,6]. Therefore, identifying accurate biomarkers and therapeutic targets for PAAD is urgently needed. Studies of RNA-binding proteins (RBPs) have demonstrated that their dysregulation plays a key role in altering RNA metabolism in various malignant tumors, and that they are considered to be attractive targets for the occurrence and aggressiveness of PAAD [39,40].

In the previous study, we explored the hub RBPs of PAAD from the TCGA and GTEx databases. Furthermore, we obtained a high-value predictive model with five RBPs. Based on the analyses from univariate Cox regressions and the Kaplan–Meier plotter database, we observed a correlation between *STAU2* expression and PAAD progression. First, we observed that the highest rate of genetic alternation in *STAU2* occurred in patients with “amplification” as the primary type, and patients with genetic mutation of *STAU2* showed poor overall survival. Therefore, a multivariate stepwise Cox regression analysis, ROC analysis, and experimental *STAU2* study showed that high expression of *STAU2* was associated with poor clinical outcomes, and that downregulation of *STAU2* resulted in a decrease in the growth, migration, and invasion of PAAD cells and induced apoptosis. Moreover, after analyzing the overlap of STAU2-CLIP, *STAU2*-correlated genes, and *STAU2*-correlated downregulated genes in sh*STAU2* samples, we identified *PALLD*, *HNRNPU*, *SERBP1*, and *DDX3X* as *STAU2* target genes. Furthermore, increased *STAU2* expression correlated positively with immune cell PAAD infiltration and immune checkpoint expression. Our data also suggested that *STAU2* expression induces immunosuppression through accumulation of Treg, TAM, and MDSC. Finally, the analyses of the GDSC and CTRP databases indicated that *STAU2* expression was associated with drugs resistance, and that patients with high *STAU2* were more sensitive to chemotherapy drugs (5-Fluorouracil and Gemcitabine) but more resistant to Erlotinib, an EGFR inhibitor.

Based on the data from this study, we demonstrated that the RNA-binding protein *STAU2* is a useful regulator of PAAD initiation and progression, suggesting that targeting RBPs is a promising therapeutic strategy for patients with PAAD. Dysregulation of RNA metabolism by altering RBP expression was associated with PAAD occurrence and aggressiveness. Therefore, investigating the most commonly suitable RBPs may lead to promising innovative therapy targets. In the present study, based on the TCGA and GTEx databases, we identified 477 DERBPs and screened out five hub RBPs (*STAU2*, *DDX60L*, *MRPS10*, *PARN*, *TLR3*) to build our prognostic signature. By adopting univariate and multiple stepwise Cox and lasso regression analyses in TCGA–PAAD, we found the profiles of these five genes to be overexpressed in PAAD samples. In addition, the TCGA cohort KM survival map showed that high expression of these five genes was associated with poorer overall survival. Furthermore, high expression of *STAU2*, *DDX60L*, *MRPS10*, and *PARN* was also evaluated by a KM survival plot through ICGC–PAAD. The results collectively demonstrated that the five hub RBP-derived risk model demonstrated a high prognostic and diagnostic ability.

DEGs in different risk groups were used to study the molecular mechanisms of patients at high and low risk for PAAD. In total, 3177 downregulated and 741 upregulated genes were screened out, and Hallmark and KEGG functional enrichments were performed. We found that pancreatic secretion, E2F targets, G2M targets, MTORC1 signaling, and glycolysis signaling were highly activated in the high-risk group. These results indicated that these RBPs might regulate cell proliferation and PAAD metabolism pathways. Several studies have demonstrated the function of these genes in various cancers. For instance, Bajaj et al. reported that *STAU2* is a critical factor in the development of myeloid leukemia because it drives histone methylation [34]. DExD/H-Box 60 (DDX60L), a member of the DExD/H-Box family of helicases involved in RNA metabolism, has been identified as influencing the survival and metastasis of pancreatic ductal adenocarcinoma (PDAC) cells [41]. Mitochondrial ribosomal protein S10 (MRPS10), a 28S subunit protein belonging to the S10P ribosomal protein family, was found to be elevated in breast cancer, which might promote the fatty acid oxidation (FAO) process to support the rapid metabolism of tumor cells [42]. Toll-like receptor 3 (TLR3) is an important member of the TLR family and is involved in double-stranded RNA binding and activation of the NF-κB signaling pathway. It has been reported as an oncogene involved in the proliferation of tumor cells that are highly expressed in head and neck, prostate, and breast cancers, as well as in hepatocellular carcinoma and multiple myeloma [43]. Poly(A)-specific ribonuclease (PARN), which removes adenosine residues from the poly(A) tails after catalyzing mRNA deadenylation, was upregulated in gastric cancer, acute leukemia, and small cell lung carcinoma [44]. Furthermore, we found that the genetic alteration of *STAU2*, *MRPS10*, and *PARN* correlated with mRNA levels, and that the mRNA level of *STAU2* was also regulated by methylation. In addition, *DDX60L* and *TLR3* might be methylation-regulating genes.

After focusing on the expression, hazard ratios with prognostic values, and survival tests of these five hub genes, we turned our attention to *STAU2* to evaluate the reliability of the RBP-related signature. *STAU2* contains five conserved RNA-binding domains that stabilize and transport mRNA. While it has been studied in hematological malignancies, its function in PAAD remains unknown. Herein, we showed that *STAU2* displays high genetic alteration frequency among multiple tumors in TCGA cohorts, and that genetically mutated *STAU2* patients showed poor prognosis in overall survival. Furthermore, multivariate Cox regression analysis showed that in the TCGA and ICGC databases, high expression of *STAU2* mRNA was an independent prognostic factor for PAAD patients, and ROC analysis predicted that *STAU2* expression was associated with the progression of PAAD. In addition, we confirmed that the transcript and protein levels of STAU2 are higher in PAAD cells than in normal pancreatic cells (HPDE6-C7). We observed that silencing *STAU2* not only inhibited cell proliferation, colony formation, and metastasis but also induced tumor apoptosis. These results suggest that *STAU2* plays a key role in PAAD regulation and is a potential target for the development of new anticancer drugs.

Notably, we explored co-expression genes with *STAU2* using the Linkedomics dataset, interactive genes using the STRING database, and relative genes using the FPKM matrix across TCGA–PAAD patients. Hallmark and KEGG functional analyses revealed that they were enriched mainly in processes such as UV-response, mitotic spindle, RNA transport, the mRNA surveillance pathway, TGF-β signaling, and the inflammatory response pathway. Furthermore, we identified seven key target genes of *STAU2* by exploring the *STAU2*-correlated genes (TCGA–PAAD), *STAU2*-binding genes (GSE134971), and downregulating *STAU2*-correlated genes (RNA-Seq) database. Correlation analysis and RT-qPCR and RIP RT-qPCR validation showed that *STAU2* binds and stabilizes *PALLD*, *HNRNPU*, *SERBP1*, and *DDX3X* mRNA, and that upregulation of these genes leads to poor overall survival in PAAD patients. Previous studies have reported the biological function of these targets in several kinds of cancers. PALLD promotes pancreatic cancer cell invasion by promoting the invasive formation of tumor-associated fibroblasts [45]. HNRNPU is overexpressed in hepatocellular carcinoma (HCC) and promotes HCC occurrence and progression [46,47]. SERBP1 is a member of the RG/RGG family of RBPs, which is markedly overexpressed in glioblastoma and in prostate, ovarian, and liver cancer and is associated with poor outcomes [48,49]. DDX3X is a member of the ATP-dependent RNA helicase subfamily, which is involved in mRNA splicing, transport, and translation. DDX3X functions as an oncogenic protein in regulating the tumorigenesis and metastasis of various cancers, such as glioma, prostate cancer, Ewing sarcoma, and breast cancer [50,51]. Moreover, Liang et al. reported that upregulation of DDX3X is associated with a poor prognosis in pancreatic ductal adenocarcinoma (PDAC) patients [52]. The molecular regulatory mechanism between *STAU2* and these substrates needs to be explored further.

In recent years, immunotherapy has been applied in a variety of cancers, but due to the heterogeneity of tumors and complexity of the tumor microenvironment, PAAD patients display poor response to single-agent immunotherapy [2]. Immune infiltration levels exhibit a close relation to immunotherapy responses and exert an important role in affecting the prognosis of patients. Here, we found that *STAU2* correlated positively with the common lymphoid progenitor, granulocyte–monocyte progenitors, CD8+ T cell, CD8+ central memory T cell, and CD4+ memory T cell. Meanwhile, *STAU2* negatively correlated with B plasma, CD4+ central memory, and CD4+ Th1 T cells in PAAD. Moreover, *STAU2* expression was positively associated with immune checkpoint markers, such as PD-1, PD-L1, and CTLA4. Our study also showed that *STAU2* expression was positively associated with marker sets of immunosuppressive cells (Treg, TAM, and MDSC). More importantly, PAAD patients with high *STAU2* expression and decreased CD8+ T cells, eosinophils, macrophages, regulatory T cells, and type 2 T-helper cells exhibited poor prognoses. Collectively, these results indicate that a high *STAU2* expression might contribute to PAAD immune system evasion, and this could serve as a potential immunotherapeutic target. Currently, the main treatment strategies are surgery and chemotherapy. Previous studies have shown that RBPs are closely related to drug sensitivity [53,54]. Our results showed that patients with high *STAU2* mRNA were more sensitive to 5-Fluorouracil and Gemcitabine, but more resistant to Erlotinib (an EGFR inhibitor), which could serve as a potential target for combinatorial therapy.

## 5. Conclusions

In summary, our study carried out a risk model based on five RBPs via a series of bioinformatic techniques in PAAD–TCGA and GTEx datasets. High expression of these five RBPs offers excellent prognostic and diagnostic potential. Among these genes, *STAU2* was the most high-potential biomarker, involved in PAAD occurrence and progression by regulating the substrate mRNA surveillance pathway, which was confirmed by the experiment. Furthermore, we found that a high expression level of *STAU2* not only contributes to PAAD immune system evasion but also correlates with chemotherapy drug sensitivity, which implies that *STAU2* could serve as a potential target for combinatorial therapy. All these findings indicate that *STAU2* is a novel prognostic and diagnostic biomarker for PAAD, which highlights the attractive potential of RBPs in cancer therapy and drug development.

## Figures and Tables

**Figure 1 cancers-14-03629-f001:**
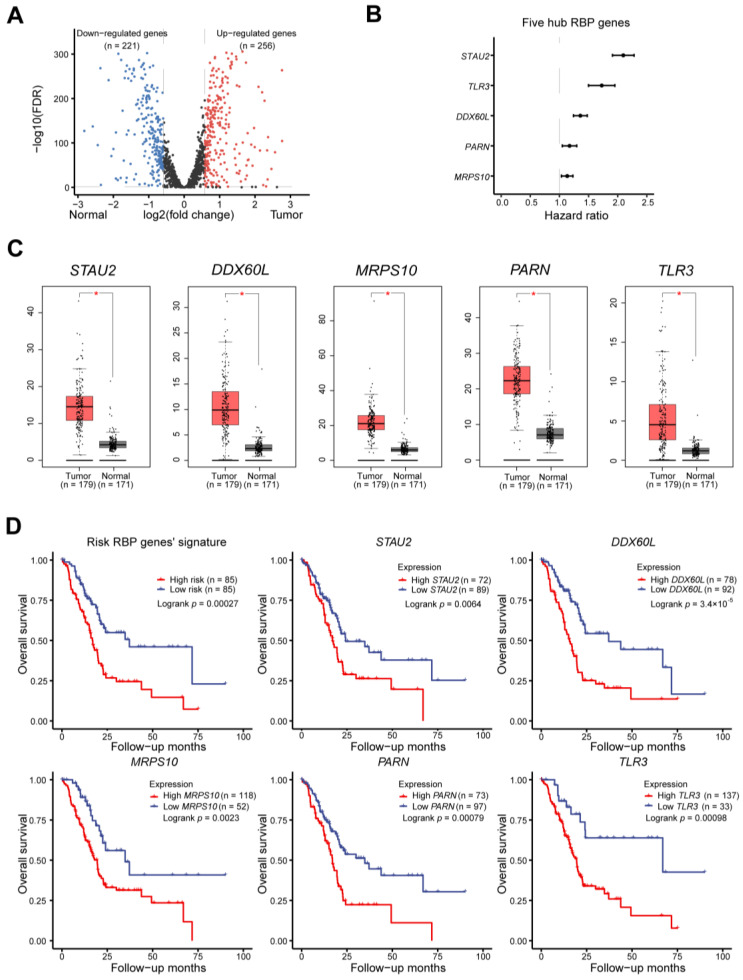
Identification of prognostic RBP signature. (**A**) Volcano plot shows that 477 RBP genes are significantly up- and down-regulated in PAAD patients. (**B**) Forest plots to show hazard ratios of the hub RBP genes, with prognostic values in PAAD based on data from TCGA. (**C**) Boxplots show a comparison of expression levels in TCGA tumor samples and GTEx normal tissues. The *p*-values were calculated using the unpaired Mann–Whitney Wilcoxon test, * *p* < 0.01. (**D**) The Kaplan–Meier plotter shows the relationship between RBP gene expression levels and overall survival in six prognostic PAAD patients.

**Figure 2 cancers-14-03629-f002:**
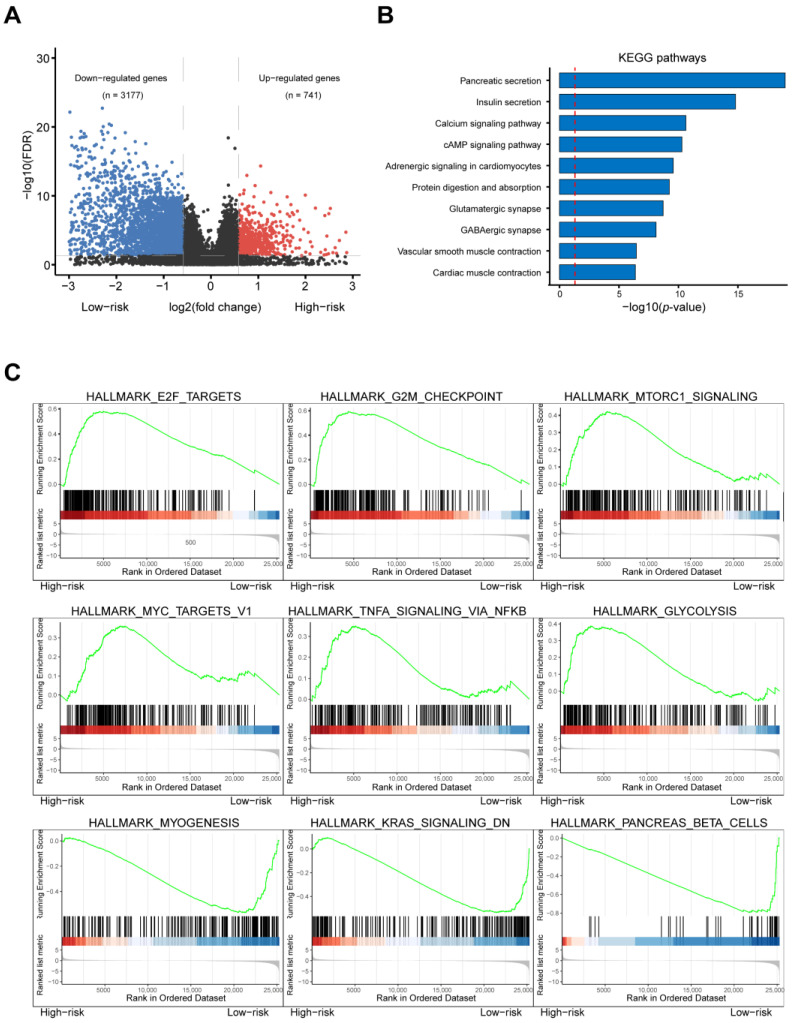
Transcriptomic alternation in high-risk PAAD patients. (**A**) The Volcano plot shows significant differences in gene expression between PAAD high-risk subgroups and low-risk subgroups. (**B**) Bar graphs show KEGG pathway enrichment for risk-related DEGs in PAAD patients. (**C**) Diagram of canonical gene set enrichment analysis clearly shows up- and down-regulated hallmark pathways in high-risk subgroups.

**Figure 3 cancers-14-03629-f003:**
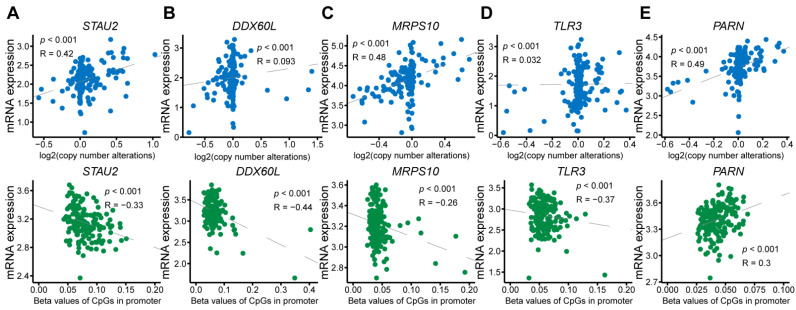
Upstream regulation of alternated hub RBPs in PAAD patients. Top scatter plots showed the relationship between mRNA expression and copy number alterations in (**A**) *STAU2*, (**B**) *DDX60L*, (**C**) *MRPS*, and (**D**) *TLR3*. (**E**) The bottom scatterplot of *PARN* shows the relationship between mRNA expression and promoter DNA methylation levels in (**A**) *STAU2*, (**B**) *DDX60L*, (**C**) *MRPS*, (**D**) *TLR3*, and (**E**) *PARN*.

**Figure 4 cancers-14-03629-f004:**
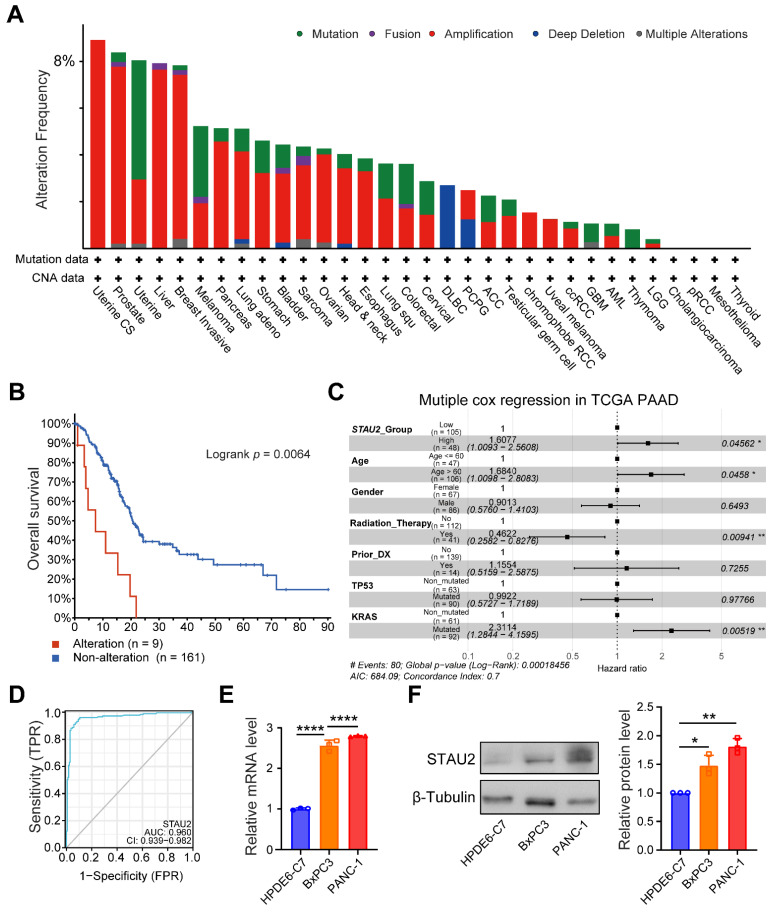
High expression of *STAU2* correlates with prognosis in PAAD. (**A**) Bar plots to show the genetic alteration frequency of *STAU2* among multiple tumors in TCGA. (**B**) Kaplan–Meier plots show overall survival in *STAU2* alteration or non-alteration patients. (**C**) Forest plot showing the hazard ratios of multivariate Cox regression of TCGA in PAAD patients. (**D**) The receiver operating characteristic curve (ROC) predicts the accuracy of *STAU2* as a diagnostic factor for pancreatic adenocarcinoma. (**E**) RT-qPCR experiment to show expression of *STAU2* in HPDE6-C7, BXPC-3, and PANC-1. Statistical analysis of the results of three independent experiments was performed using the unpaired Student’s *t*-tests: **** *p* <0.0001. Error bar, mean ± SD, *n* = 3. (**F**) The protein levels of STAU2 in HPDE6-C7, BXPC-3, and PANC-1 cells were detected by western blot. Grayscale analysis of the western blot results. Statistical analysis of the results of three independent experiments was performed using the unpaired Student’s *t*-tests: * *p* < 0.05, ** *p* < 0.01. Error bar, mean ± SD, *n* = 3. The uncropped Western Blot images can be found in Appendix A.

**Figure 5 cancers-14-03629-f005:**
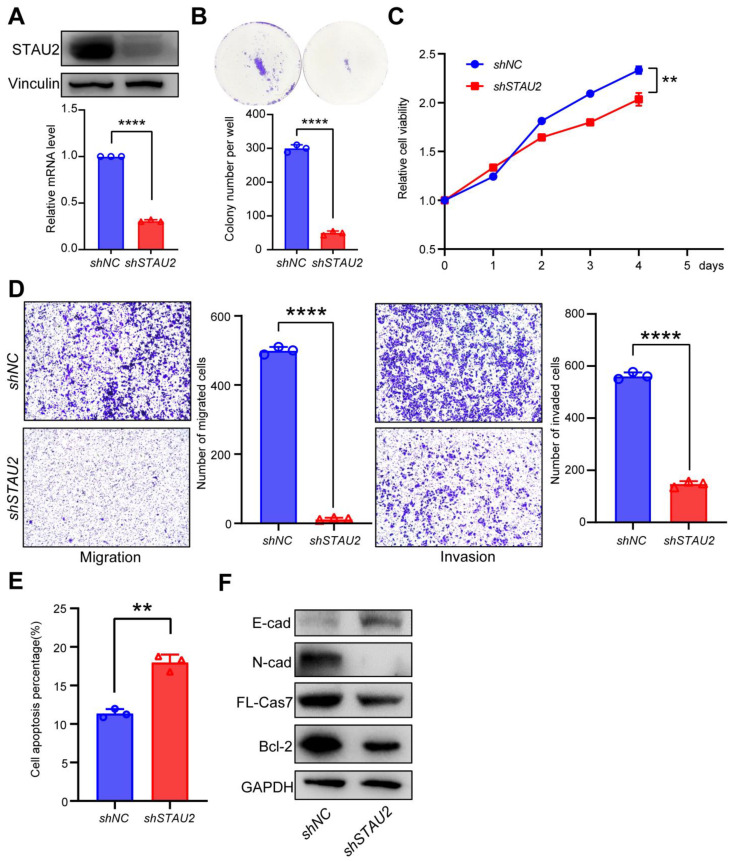
Functional analysis of *STAU2* gene downregulation in pancreatic cancer. (**A**) Western blot analysis of STAU2 protein levels in vector-transfected *shSTAU2* or PANC-1 cells. Vinculin was used as an endogenous control. The uncropped Western Blot images can be found in Appendix A. (**B**–**D**) The effect of knocking down the *STAU2* gene on colony formation (**B**), cell growth (**C**), migration (**D**), left, cells stained with crystal violet were cells migrating to the low surface), and invasion (**D**), right, crystal violet stained cells invading through Matrigel to the low surface) in PANC-1 cell lines. (**E**) Representative flow cytometry plot for quantifying apoptosis in PANC-1 *STAU2* KD and control cells. (**F**) Western blot analysis of protein changes associated with cell migration, invasion, and apoptosis following STAU2 knock-down. Statistical analysis of the results of the above three independent experiments was performed using the unpaired Student’s *t*-tests: ns *p* > 0.05, ** *p* < 0.01, **** *p* <0.0001. Error Bar, mean ± SD, *n* = 3. The uncropped Western Blot images can be found in Appendix A.

**Figure 6 cancers-14-03629-f006:**
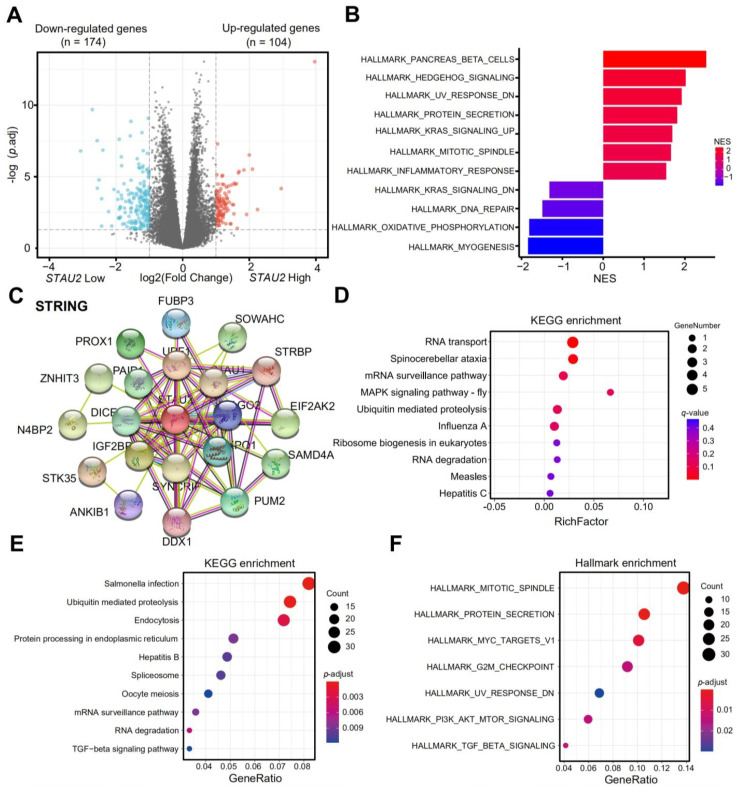
The function analysis of *STAU2* and its-related genes. (**A**) Volcano plots show significantly differentially expressed genes between high and low expressing subpopulations of *STAU2* in PAAD. (**B**) GSEA enrichment of the *STAU2* correlated genes. (**C**) The PPI analysis of STAU2 (STRING). (**D**) KEGG enrichment of the PPI genes. (**E**) Dot plots showing enriched KEGG pathways of *STAU2*-related genes. (**F**) Dot plots showing hallmark pathway enrichment results of *STAU2*-related genes.

**Figure 7 cancers-14-03629-f007:**
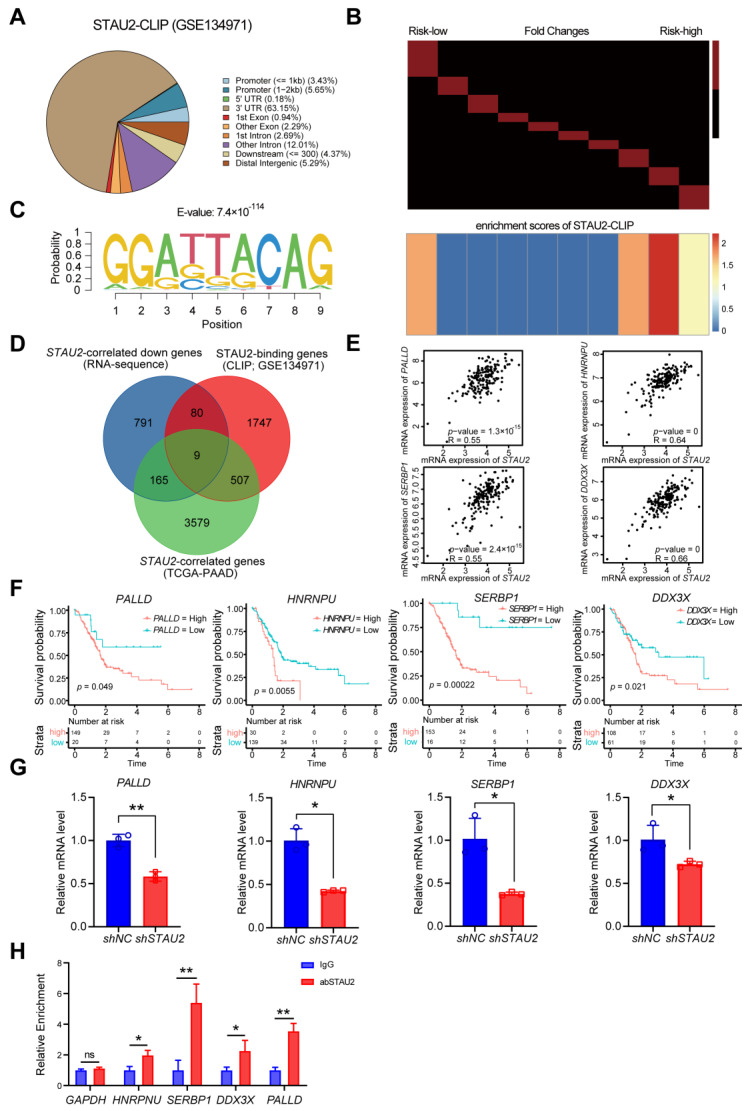
Identification of STAU2-target genes. (**A**) The genomic distribution of STAU2-CLIP peaks (GSE134971), presented by Venn diagram. (**B**) Heatmaps to show the ordered genes based on their fold-changes between low-risk and high-risk PAAD patients. These genes were then grouped into the same fraction (approximately 1000 genes per expression compartment). The red bar on a black background indicates the range of values for each container (the minimum value is −2, and the maximum value is 2). The corresponding heatmap below shows the enrichment scores of STAU2; red or yellow represent the overrepresentation of the STAU2 binding motif, while blue represents the underrepresentation of the STAU2 binding motif. (**C**) Logo representation plots to show the putative STAU2 binding motifs overrepresenting around STAU2-binding genes. (**D**) Venn diagrams show the overlap of *STAU2*-correlated genes, *STAU2*-binding genes, and *STAU2*-correlated down genes. (**E**) Scatter plots to show the correlation between *STAU2* and STAU2-target genes. (**F**). Kaplan–Meier plot showing the relationship between expression levels of STAU2-target genes and overall survival of TCGA–PAAD patients. (**G**) RT-qPCR experiments demonstrated the expression of STAU2 target genes in PANC-1 cells transfected with sh*STAU2* or vector. Statistical analysis was performed on the results of three independent experiments using the *t*-test: ns *p* > 0.05, * *p* < 0.05. Error bar, mean ± SD, *n* = 3. (**H**) Relative enrichment of STAU2 target genes in PANC-1-abSTAU2 over IgG (control) determined by RIP RT-qPCR assays. GAPDH shown as negative control. Data were generated from three independent trials, Statistical analysis was performed using the *t*-test: * *p* < 0.05, ** *p* < 0.01. Error bar, mean ± SD, *n* = 3.

**Figure 8 cancers-14-03629-f008:**
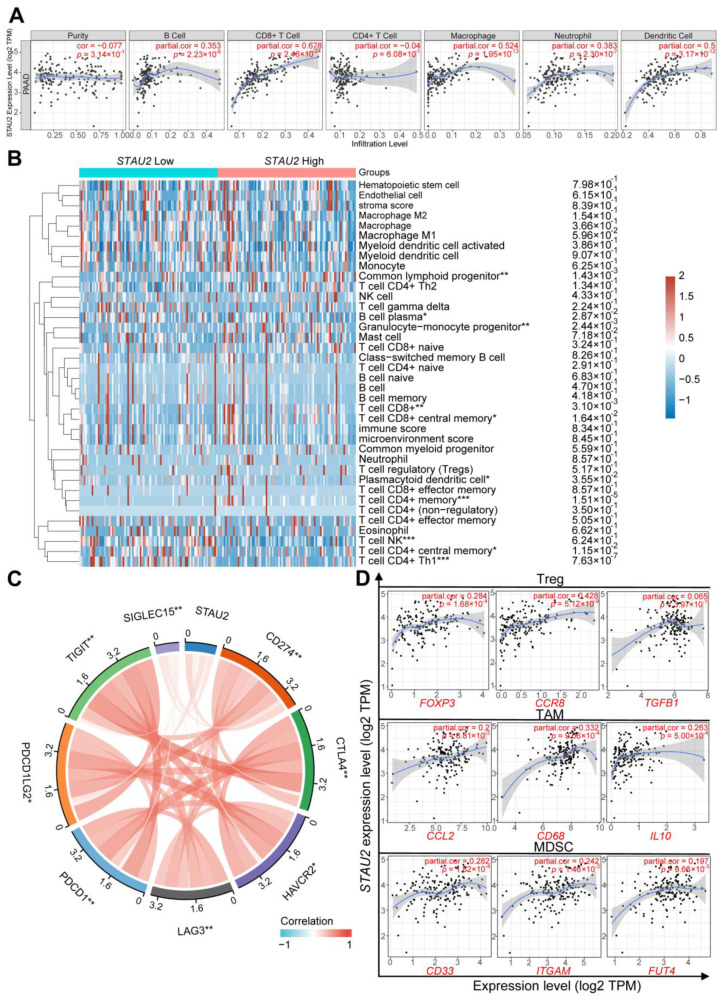
Correlation analysis of *STAU2* expression with levels of immune infiltration and immune checkpoints in pancreatic adenocarcinoma (**A**) *STAU2* positively correlates with infiltration of B cells, CD8+ T cells, macrophages, neutrophils, and dendritic cells by the TIMER. (**B**) Immune cells’ score heatmap associated with *STAU2* expression. * *p*  <  0.05, ** *p*  <  0.01, *** *p*  <  0.001. (**C**) Correlation analysis of *STAU2* expression level and the level of eight immune control points in pancreatic cancer * *p  *<  0.05, ** *p*  <  0.01. (**D**) Correlation analysis of *STAU2* and Treg, TAM, and MASC cell gene markers in TIMER.

**Figure 9 cancers-14-03629-f009:**
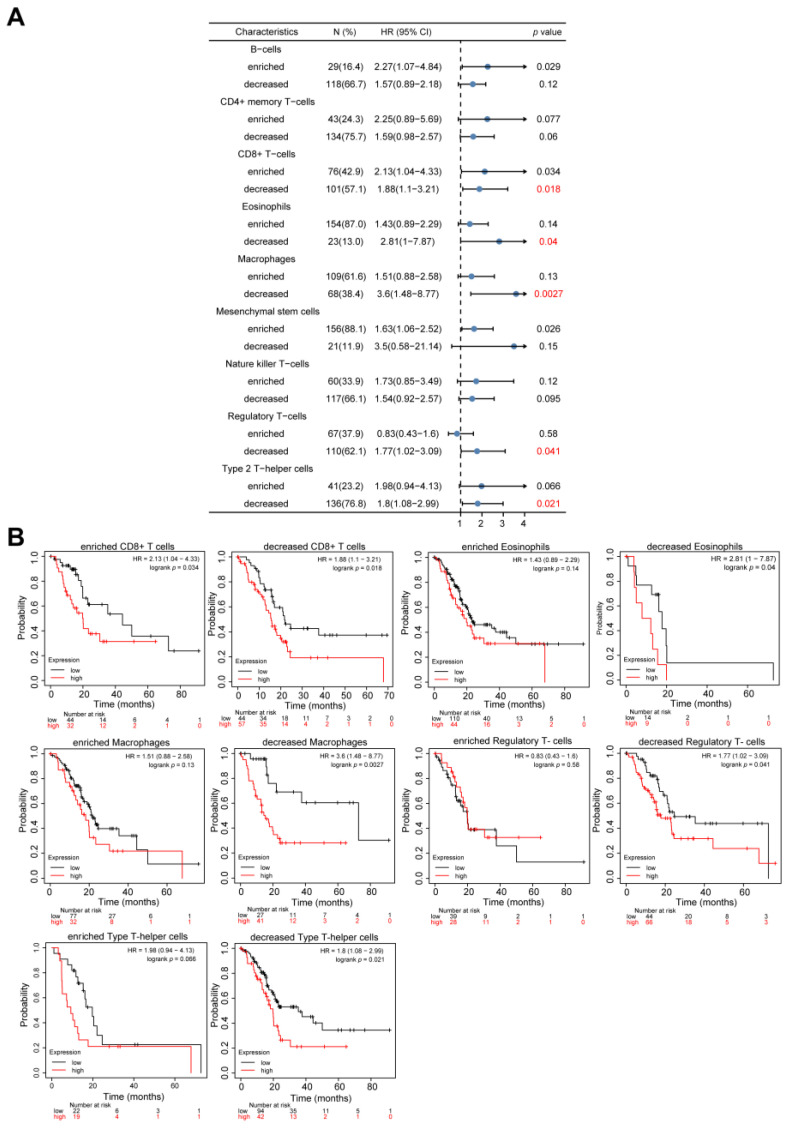
Kaplan–Meier survival curve of high and low expression of *STAU2* in pancreatic cancer-immune cell subsets. (**A**) Forest plot showing the prognostic value of *STAU2* expression in different subsets of immune cells in PAAD patients. (**B**) The Kaplan–Meier plotter method was used to demonstrate the correlation between *STAU2* expression and OS in different subsets of immune cells from PAAD patients.

**Figure 10 cancers-14-03629-f010:**
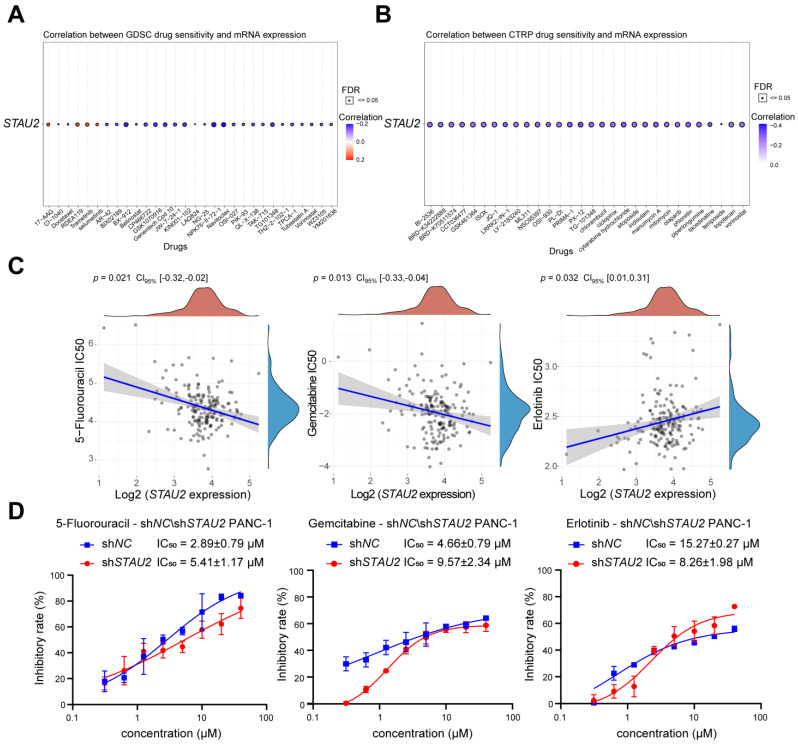
Prediction of the relationship between *STAU2* expression and drug sensitivity. (**A**) Drug sensitivity of *STAU2* from GDSC. Dot plots show correlations between gene expression and GDSC connectivity. (**B**) Drug sensitivity of *STAU2* from CTRP. Scatter plot showing the correlation between gene expression and CTRP binding. The size of the dots represents the FDR value, and the color of the dots represents the correlation, with red being positively correlated and blue being negatively correlated. (**C**) The correlation between IC_50_ score of first-line drugs for pancreatic adenocarcinoma and *STAU2* gene expression was analyzed by Spearman correlation. (**D**) IC_50_ of sh*STAU2* PANC-1 cell and sh*NC* PANC-1 cell treated by 5-Fluorouracil, Gemcitabine, and Erlotinib. Data were generated from three independent trials. Error bar, mean ± SD, *n* = 3.

## Data Availability

The data that support the findings of this study are available in the Appendix A of this article.

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
