# Peer review of "Systematic Identification of the RNA-Binding Protein STAU2 as a Key Regulator of Pancreatic Adenocarcinoma"

_cancers, 2022, doi:10.3390/cancers14153629_

Round 1
Reviewer 1 Report
The authors have addressed all my questions.
Author Response
We would like to express our sincere thanks to reviewers for the positive comments and professional suggests!
Reviewer #1:
The authors have addressed all my questions.
Response: Thank you very much for your help and recognition of our manuscript!
Reviewer 2 Report
In this manuscript, the authors identified STAU2 as one of hub RBPs in PAAD and characterized its potential functions and underling mechanisms in PAAD aggression. By data mining in TCGA and GTEx, the authors found STAU2 could be a prognostic RBP signature of PAAD and was also associated with high-risk PAAD patients. Mechanically, they demonstrated that downregulation of STAU2 resulted in reduction of growth, invasion and migration of cancer cells. They also showed that STAU2 related to immune escaping and chemotherapy drug sensitivity, suggesting its therapeutic potential for PAAD. In general, this study is well designed and the data is scientifically sound. There are several issues that need to clarify.
1, The STAU2-CLIP data used in this study is actually from leukemia cells and therefore is not suitable to identify STAU2 targets in PAAD. The authors should generate STAU2-CLIP data in PAAD cells and compare these data with the data from leukemia cells to further investigate the regulation specificity of STAU2.
2, In figure 7H, the author should add some gene loci as negative control to show similar STAU2 enrichment level in IgG and abSTAU2.
Author Response
We would like to express our sincere thanks to reviewers for the positive comments and professional suggests!
Reviewer #2:
In this manuscript, the authors identified STAU2 as one of hub RBPs in PAAD and characterized its potential functions and underling mechanisms in PAAD aggression. By data mining in TCGA and GTEx, the authors found STAU2 could be a prognostic RBP signature of PAAD and was also associated with high-risk PAAD patients. Mechanically, they demonstrated that downregulation of STAU2 resulted in reduction of growth, invasion and migration of cancer cells. They also showed that STAU2 related to immune escaping and chemotherapy drug sensitivity, suggesting its therapeutic potential for PAAD. In general, this study is well designed and the data is scientifically sound. There are several issues that need to clarify.
Response: Thank you for your positive review of the importance of our work!
1, The STAU2-CLIP data used in this study is actually from leukemia cells and therefore is not suitable to identify STAU2 targets in PAAD. The authors should generate STAU2-CLIP data in PAAD cells and compare these data with the data from leukemia cells to further investigate the regulation specificity of STAU2.
Response: Thank you very much for the professional suggestion! We will definitely detect STAU2-CLIP in PAAD in the future. In order to further verify downstream of STAU2 targets, we used STAU2 downstream target genes in CML and intersections them with genes negatively associated with STAU2 expression in pancreatic cancer TCGA database. We selected nine genes based on their association with function in cancer, and RIP RT-qPCR validation was performed. Seven of these genes were identified as STAU2 targets in PAAD cells.
2, In figure 7H, the author should add some gene loci as negative control to show similar STAU2 enrichment level in IgG and abSTAU2.
Response: Thank you very much for your great advice! We have modified the manuscript according to your suggestion in figure 7H.
Figure 7. Identification of STAU2-target genes. (H). Relative enrichment of STAU2 target genes in PANC-1-abSTAU2 over IgG (control) determined by RIP RT-qPCR assays. GAPDH shown as negative control.

Reviewer 3 Report
In this manuscript the authors identified 5 RNA binding proteins using bioinformatic analysis to assess the prognosis of the pancreatic adenocarcinoma. STAU2 was selected for further studies to demonstrate its abundance in pancreatic cancer that co-relates with poor survival of the patients. Interestingly, the authors demonstrated that pancreatic cancer cells are sensitive to 5FU and Gemcitabine in relation to STAU2 making it a potential therapeutic drug target. Overall, the authors complied with the comments of the reviewers and made significant improvements in the manuscript. The font sizes in three of the figures in the manuscript could be increased to make it easily readable. Here are my minor comments:
1. Figure 4C could be addressed by increasing the font size of the words in the columns and rows.
2. Figure 8A & 8D could be addressed by increasing the font size of the words in the figures.
Author Response
We would like to express our sincere thanks to reviewer for the positive comments and professional suggests!
Reviewer #3:
In this manuscript the authors identified 5 RNA binding proteins using bioinformatic analysis to assess the prognosis of the pancreatic adenocarcinoma. STAU2 was selected for further studies to demonstrate its abundance in pancreatic cancer that co-relates with poor survival of the patients. Interestingly, the authors demonstrated that pancreatic cancer cells are sensitive to 5FU and Gemcitabine in relation to STAU2 making it a potential therapeutic drug target. Overall, the authors complied with the comments of the reviewers and made significant improvements in the manuscript. The font sizes in three of the figures in the manuscript could be increased to make it easily readable. Here are my minor comments:
Response: Thank you very much for positive comments on the importance of our work! This reflects the advantages of our manuscript!
1.Figure 4C could be addressed by increasing the font size of the words in the columns and rows.
Response: Thank you very much for your great advice! We have modified the manuscript according to your suggestion.
- Figure 8A & 8D could be addressed by increasing the font size of the words in the figures.
Response: Thank you very much for your great advice! We have modified the manuscript according to your suggestion.
Reviewer 4 Report
The manuscript ID entitled "Systematic Identification of the RNA-Binding Protein STAU2 2 as a Key Regulator of Pancreatic Adenocarcinoma" is written well.
However, the following points need to be addressed, the sample size is low for bioinformatic analysis alone. Further, the elucidated marker whether it is patient-specific (i.e.geographical) or clinical features specific gives more credit to the present research.
Author Response
We would like to express our sincere thanks to reviewer for the positive comments and professional suggests!
Reviewer #4:
The manuscript ID entitled "Systematic Identification of the RNA-Binding Protein STAU2 2 as a Key Regulator of Pancreatic Adenocarcinoma" is written well.
Response: Thank you very much for your positive commons on this manuscript!
However, the following points need to be addressed, the sample size is low for bioinformatic analysis alone. Further, the elucidated marker whether it is patient-specific (i.e.geographical) or clinical features specific gives more credit to the present research.
Response: Thank you very much for this scientific comment! The pancreatic cancer data information and sample size we used were obtained from TCGA database and ICGC database. For patient specificity (geographic specificity) and clinical features specificity of marker, we respectively conducted expression and prognosis analysis in different conditions. For the geographical specificity of patients, we analyzed STAU2 expression in pancreatic cancer samples of different ethnicities (White, Asian and Black) and found no significant differences. For clinical features specificity, we have performed multivariate Cox regression analysis of pancreatic cancer prognosis in Figure 4C and Figure S3 and found that age, radiotherapy, and KRAS mutation had a significant prognostic impact. And we analyzed STAU2 expression in pancreatic cancer patients without and with radiation therapy, and found that pancreatic cancer patients who received radiation therapy had significantly lower STAU2 expression than those who did not. We also experimentally verified that patients with high STAU2 mRNA levels were more sensitive to chemotherapy drugs, such as 5-fluorouracil and gemcitabine, indicating that STAU2 is an excellent target for clinical diagnosis and therapy.

Round 2
Reviewer 2 Report
My concerns have been addressed.
Author Response
We would like to express our sincere thanks to reviewer for the positive comments and professional suggests!
Reviewer #2:
My concerns have been addressed.
Response: Thank you very much for your help and recognition of our manuscript!

Reviewer 4 Report
Thanks for the revision. However, the answers and figures in the author’s response did not incorporate in the revised manuscript. Please make sure the inclusion of the amendments.
Author Response
We would like to express our sincere thanks to reviewer for the positive comments and professional suggests!
Reviewer #4:
Thanks for the revision. However, the answers and figures in the author’s response did not incorporate in the revised manuscript. Please make sure the inclusion of the amendments.
Response: Thank you very much for your great advice! We have modified the manuscript according to your professional suggestion.
Line 412-419: For patient specificity (geographic specificity) and clinical features specificity of STAU2, we respectively conducted expression and prognosis analysis in different conditions. For the geographical specificity of patients, we analyzed STAU2 expression in pancreatic cancer samples of different ethnicities (White, Asian and Black) and found no significant differences. And we also analyzed STAU2 expression in pancreatic cancer patients without and with radiation therapy, and found that pancreatic cancer patients who received radiation therapy had significantly lower STAU2 expression than those who did not (Figure S3A and Figure S3B).

Round 3
Reviewer 4 Report
Accept in present form
This manuscript is a resubmission of an earlier submission. The following is a list of the peer review reports and author responses from that submission.
Round 1
Reviewer 1 Report
In this manuscript the authors carry out bioinformatic analyses on publicly available data derived from human pancreatic adenocarcinoma to identify RNA binding proteins associated with poor prognosis. They identified a signature of 5 RNA binding proteins, and focused the remainder of the manuscript on STAU2. Both high levels of STAU2 mRNA expression and seemingly paradoxically STAU2 mutation are associated with worse prognosis in PAAD. shRNA knockdown of STAU2 in 2 pancreatic cancer cell lines lead to a decrease in viability, mobility & invasion, while increasing the fraction of apoptotic cells.
Major Points:
1) The authors describe a statistically poorer prognosis for both STAU2 amplified/highly-expressing PAAD patients, and seemingly paradoxically poorer prognosis in STAU2 mutated tumors. The potential mechanisms of this are not elaborated upon and warrant further investigation/discussion.
2) In order to define the RNA targets of STAU2 the authors are cross-referencing CLIP data from an unrelated human hematopoietic cancer with RNAseq data derived from PAAD patients.
- At best this analysis is hypothesis generating. It is not sufficient to define direct targets of STAU2 in PAAD. In order to definitively identify relevant RNA targets in PAAD, CLIP data must be generated in the relevant cell line/cancer tissue types, and corresponding RNAseq data +/- shRNA STAU2 in cell lines investigated. This is the gold-standard in the field for defining in vivo direct protein-RNA interactions and predicting the functional outcome of such an interaction. At a minimum RNA-IP (RIP) should be carried out to show that STAU2 can bind the targets proposed in PAAP cell lines, with the caveat that RIP is subject to false positives through random re-association post-lysis, and indirect pull down in the presence of crosslinking agents.
- The additional filter of requiring protein-protein interactions between STAU2 and transcripts identified in the current analysis needs further explanation as this is not a standard requisite for defining functional RNA binding protein-target interactions.
- The data presented in Figure 4B-C is not well described and it is not clear what information is being imparted.
3) Drug sensitivity analysis of STAU2
- The derivation of the data presented in Figure 10A and B regarding the potential "drug sensitivity of STAU2" needs further clarification in the materials and methods and results section. The figures themselves are difficult to interpret due to size of dots and colors used.
- These claims would be further supported through in vitro verification in PAAD cell lines +/- shRNA knockdown of STAU2,
4) The methods section needs considerable editing for clarity and reproducibility of "wet lab" experiments.
- The number of replicates for each cell culture based experiment needs to be clearly stated (proliferation assay, rt-QPCR, Western Blot quantification, colony formation, migration/invasion, apoptosis).
- Kits and instruments used should be clearly cited.
- The plasmids used for lentivirus production should be given, as should the sequences or source of the non-targeting shRNA plasmid which is mentioned in figures but not the methods section.
- The difference between migration and invasion assay should be more clearly outlined.
5) The discussion section needs substantial editing to remove the unnecessary summarization of the results section.
This manuscript needs extensive editing for language use.
Reviewer 2 Report
In their manuscript, Wang et al. investigated a potential role of the RNA-binding protein (RBP) Staufen2 (Stau2) in pancreatic adenocarcinoma. Detailed bioinformatic analysis revealed high expression of Stau2 in pancreatic cancer cells associated with poor clinical outcomes. Moreover, the authors present data that Stau2 is needed for growth, migration and invasion of a pancreatic cancer cell line.
Major concerns:
Even though the authors made an effort to investigate the role of Stau2 in pancreatic cancer, a large part of their conclusions are based on predications and correlations from different databases. Experiments using a pancreatic cell line are the only functional proof they provide. Therefore, some conclusions are not sufficiently supported by the data presented and are too vague without providing any functional validation. For example, the authors talk about Stau2 target genes and refer to a published eCLIP dataset. In this paper, Stau2 eCLIP was done using K562 bcCML cells that were derived from chronic myeloid leukemia patients; cells that are fairly different from pancreatic cancer cells. Moreover, the authors present pathways that might be regulated by Stau2 using Stau2 low and high expressing cells. However, the broad dysregulation of mRNAs and also of RBPs in these cells impede conclusion on Stau2 regulated pathways without providing data from specific Stau2 knock-down experiments.
To bring this manuscript in a publishable format, I would strongly recommend to include functional data on Stau2. For example, the authors present RNA levels of Stau2 in pancreatic cancers but do not test for the protein. Moreover, they have done Stau2 protein interactomics but do not test for mRNAs that correlate with Stau2 expression. These experiments would definitely strengthen the link between Stau2 and cancer cell survival.
Minor comments:
- There are many typos throughout the manuscripts.
- Moreover, some analyses are not informative (g. fig. 2).
- What’s the difference between migration and invasion in fig. 5? Not clear to me.
- Figure legends would benefit from more detailed information regarding the presented plots. Moreover, axis labels are incomplete in some cases.
Reviewer 3 Report
In this manuscript, the authors have identified STAU2 as a key regulator of pancreatic adenocarcinoma by performing detailed bioinformatic analysis. However, there is still some missing experimental data to support their conclusion.
1, In Fig 4, the authors observed both increased mRNA and protein expression of STAU2 in BxPC3 and PANC-1 pancreatic cancer cells compared to normal pancreatic cells. Then, in Fig 10, the authors indicated that increased expression of STAU2 was associated with drug resistance. Has the author tested some of these drugs in their cell lines? Does increased STAU2 lead to high IC50? Also does STAU2 knockdown leads to sensitivity to certain drugs, including PLK inhibitors, BET inhibitor, and other compounds? It is worthy to have some experimental data to support the drug sensitivity analysis result.
2, In Fig 5, the authors claimed that knocking down STAU2 gene significantly inhibited the cloning formation, cell growth, migration, and invasion in PANC-1 cells. Does it also affect normal pancreatic cell survival and growth? Is STAU2 gene essential to all pancreatic cells or only pancreatic cancer cells?
3, the authors need to check the grammar and spelling errors more carefully, especially in Materials and Methods.
For example, Line 36, “States”, not “Stated”.
Line 37, missing “as” after “considered”.
Line 225, missing “.” at the end.
Line 228, 233, 248, 255, missing subject.
Line 234-235, “cells”, not “cell lines” “cells were inoculated at 32 Celsius degrees 1200 rpm for 90min”?
Line 251, “washes in blocking solution”? should be “washes in PBS”?
Line 273, “Cell apoptosis”? missing “rate”?
Reviewer 4 Report
In this study, using bioinformatics analysis, the authors identified five hub RBPs with high prognostic value in Pancreatic Adenocarcinoma and further investigated the potential functional role of one of these hub RBPs, STAU2 in PAAD. The authors have shown that high expression level of STAU2 is associated with poor clinical outcomes and chemotherapy drug sensitivity. In in vitro cell assay, knockdown of STAU2 results in reduced level of PAAD cell growth, migration and invasion. In addition, the authors identify several key target genes of STAU2, indicating the key functional role of STAU2 in PAAD. In general, this study demonstrates the important function of STAU2 in PAAD progression and chemotherapy sensitivity, which may provide novel biomarker and therapy for PAAD. The methods for data analysis in this study are sound and clear.
There are several typos in the manuscript:
1, In line 449, “STAU2 eclipse peaks” should be STAU2 eCLIP peaks.
2, In line 288, “adjust alse detection” should be adjust false detection.